# Polarity transitions of narrow bipolar events in thundercloud tops reaching the lower stratosphere

Feifan Liu[1,2,3], Torsten Neubert [3], Olivier Chanrion[3], Gaopeng Lu[1], Ting Wu [4], Fanchao Lyu [5], Weitao Lyu[5,6], Christoph Köhn [3], Dongshuai Li[3], Baoyou Zhu [1] ✉ & Jiuhou Lei [1,7,8] ✉

Blue corona discharges are often generated in thunderclouds penetrating into the stratosphere and are the optical manifestation of narrow bipolar events (NBEs) observed in radio signals. While their production appears to depend on convection, the cause and nature of such discharges are not well known. Here we show the observations by a lightning detection array of unusual amounts of 982 NBEs during a tropical storm on the coastline of China. NBEs of negative polarity are predominantly observed at the cloud top reaching the stratosphere, and positive NBEs are primarily at lower altitudes. We find that the dominant polarity changes with the typical time of development of thunderstorm cells, suggesting that the polarity depends on the phase of the storm cells. Furthermore, we find that the lightning jump of negative NBEs is associated with above-anvil cirrus plumes of ice crystals and water vapor in the lower stratosphere. We propose that variations in updrafts induce changes in the altitude and charge concentrations of the cloud layers, which lead to the polarity transition. Our results have implications for studies of the chemical perturbations of greenhouse gas concentrations by corona discharges at the tropopause.

Blue corona discharges are often observed within, or close to, the top of thunderclouds[1-4]. They are bursts of streamers from localized, fast (few 10 µs) emissions of blue light in cloud tops (pixies). They may extend upwards forming blue cones of light reaching 25–50 km altitude (blue starters and blue jets)[1,5,6] and even reach the ionosphere at about 80 km altitude with a flash of red light (gigantic jets)[7-9]. They are commonly produced in deep convective storms with cloud tops in the lower stratosphere[3,10]. Understanding how they are initiated and under what conditions has implications for understanding their perturbations to the concentrations of greenhouse gases such as nitrous

oxide ($N_2O$) and ozone ($O_3$) in the tropopause regions[11-13] and the influence of these natural processes on Earth's radiation balance[13-15].

Recent measurements from space suggest that blue corona discharges in the upper regions of thunderstorm clouds are closely related to high-altitude compact intracloud discharges, also named narrow bipolar events (NBEs)[16-20]. These are 10–30 µs radio pulses in wideband electric field records[21-24]. NBEs have been described as a short-channel (few 100 m) "bouncing wave" phenomenon from observations and modeling of their broadband signatures and the accompanying fine structure[25-28]. The production mechanism of

[1]CAS Key Laboratory of Geospace Environment, School of Earth and Space Sciences, University of Science and Technology of China, Hefei, China. [2]CMA-USTC Laboratory of Fengyun Remote Sensing, School of Earth and Space Sciences, University of Science and Technology of China, Hefei, China. [3]Department of Space and Earth Science and Technology, Technical University of Denmark (DTU Space), Kongens Lyngby, Denmark. [4]Department of Electrical, Electronic and Computer Engineering, Gifu University, Gifu, Japan. [5]Nanjing Joint Institute for Atmospheric Sciences, Nanjing, China. [6]State Key Laboratory of Severe Weather, Chinese Academy of Meteorological Sciences, Beijing, China. [7]Mengcheng National Geophysical Observatory, University of Science and Technology of China, Hefei, China. [8]CAS Center for Excellence in Comparative Planetology, Hefei, China. ✉e-mail: zhuby@ustc.edu.cn; leijh@ustc.edu.cn

NBEs has been associated with the so-called fast positive and negative breakdown in which streamers propagate at speeds of $10^7$–$10^8$ m/s[29–31]. The exact nature of the fast breakdown that produces NBEs and can move at >$10^7$ m/s through virgin air remains a mystery. There are recent attempts to model NBEs and explain their electromagnetic signatures that consider fast breakdown and the "rebounding" observed by Rison et al. [29] using modified transmission line models.

NBEs are classified as positive or negative according to the orientation of the associated current pulse[22,32,33]. Similar to the polarity asymmetry in normal lightning, NBE polarities exhibit distinct differences[32,34]. The sources of negative NBEs are usually observed to be close to the cloud tops, and the positive ones are deeper in the clouds[18,19,23,32]. NBEs radiate intense electromagnetic waveforms at very low/low frequencies (VLF/LF) and emit the strongest very high frequency (VHF) radio signals in nature. The VHF pulses can penetrate the ionosphere and therefore be detected both by satellites and ground receivers[35,36]. In this respect, radio observations of NBEs are now offering a new means to observe the properties and conditions for initiation of blue corona discharges[18].

Here we show a storm with a high number of 672 negative and 310 positive NBEs. The storm is at a location that allows for the detection of NBEs by a dedicated ground-based sensor network and cloud parameters by a weather radar close to the center of the electrical activity. With simultaneous observations of cloud properties from space, the data provide an unprecedented opportunity to understand how cloud properties affect the generation of discharges. We find that the dominant polarity of NBEs depends on the phase of the storm cells and propose a conceptual model of the dynamic distribution of charge regions in clouds.

## Results

### The storm and the NBEs

On 30 July 2017, the tropical storm Haitang arrived on mainland China at 18:00 UTC, with maximum wind speeds reaching 20 ms⁻¹. The infrared (IR) brightness temperature derived from the geostationary meteorological satellite Himawari-8[37] is shown in Fig. 1a with the storm track. In the rain bands close to the coast, a convective region with cloud-top temperatures as cold as 185 K and below has developed. The measurements of a weather radar located on the coast presented in Fig. 1b show that the clouds reach several kilometers into the stratosphere. The region has high lightning activity measured by the Global Lightning Detection Network (GLD360), and the domination of negative cloud-to-ground (−CG) and positive intra-cloud (+IC) discharges suggest a normally electrified storm with a main negative charge region and an upper positive region[38].

NBEs are observed by ground-based lightning detection sensors at locations marked with green triangles in Fig. 1a. An example of a negative NBE is shown in Fig. 1c. The electric field waveform is recorded by four radio stations that are stacked with distance to the NBE. The radio signal shows the typical NBE signature of negative polarity. It is characterized by a single bipolar pulse propagating directly to the receivers ($t = 0$) and two ionospheric reflection pulses. The time delay of the main pulses allows estimation of the source location and the delay between the main pulses and the reflected ones helps estimate the source height[39,40]. The estimated source altitude (19.5 km) and distance lead to the modeled reflection times shown by the solid red lines.

### Alternate dominance of negative and positive NBEs

During the storm's landfall from 16:00 UTC to 20:00 UTC, 672 negative and 310 positive NBEs were observed (see "Methods", subsection

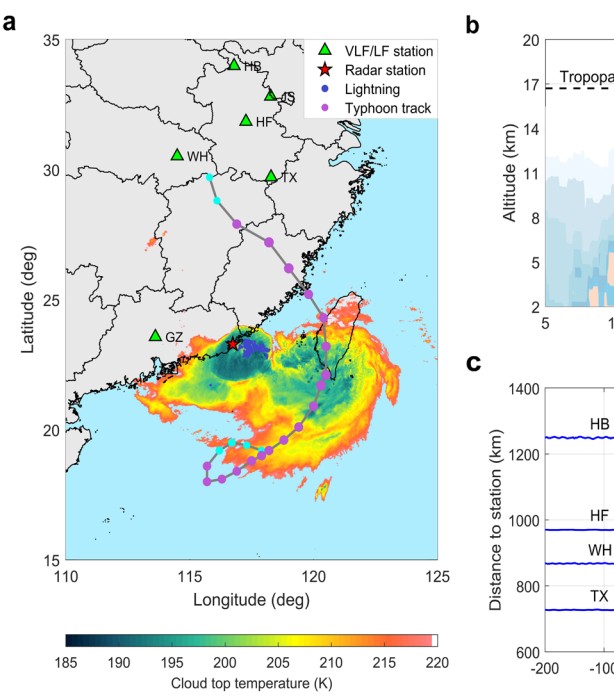

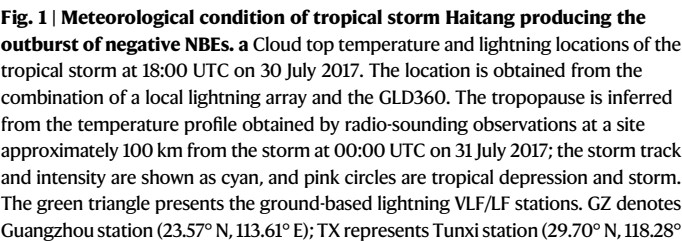

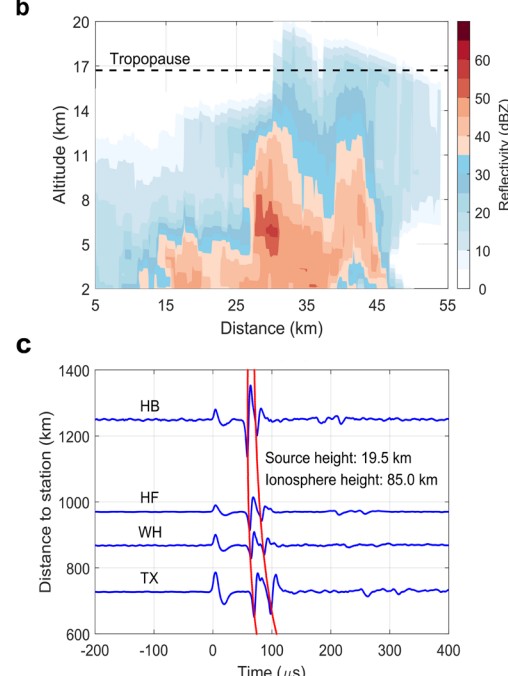

**Fig. 1 | Meteorological condition of tropical storm Haitang producing the outburst of negative NBEs. a** Cloud top temperature and lightning locations of the tropical storm at 18:00 UTC on 30 July 2017. The location is obtained from the combination of a local lightning array and the GLD360. The tropopause is inferred from the temperature profile obtained by radio-sounding observations at a site approximately 100 km from the storm at 00:00 UTC on 31 July 2017; the storm track and intensity are shown as cyan, and pink circles are tropical depression and storm. The green triangle presents the ground-based lightning VLF/LF stations. GZ denotes Guangzhou station (23.57° N, 113.61° E); TX represents Tunxi station (29.70° N, 118.28°

E); WH denotes Wuhan station (30.51° N, 114.50° E); HF, HB, and JS stand for Hefei station (31.84° N,117.27° E), Huaibei station (33.98° N,116.91° E), and Jiashan station (32.81° N,118.27° E), respectively. **b** Vertical profile of radar reflectivity along 23.25° N latitude at 18:00 UTC. The S-band radar observation is obtained from the Shantou station (23.24° N, 116.89° E) approximately 100 km away from the storm (see "Methods"). **c** A negative NBE radio signal was recorded by four distant VLF/LF stations. The solid red line presents the arrival of the estimated ionospheric reflection waveform versus distance assuming that the reflection height of the ionosphere D layer is at 85 km altitude and that the source height of NBEs is 19.5 km.

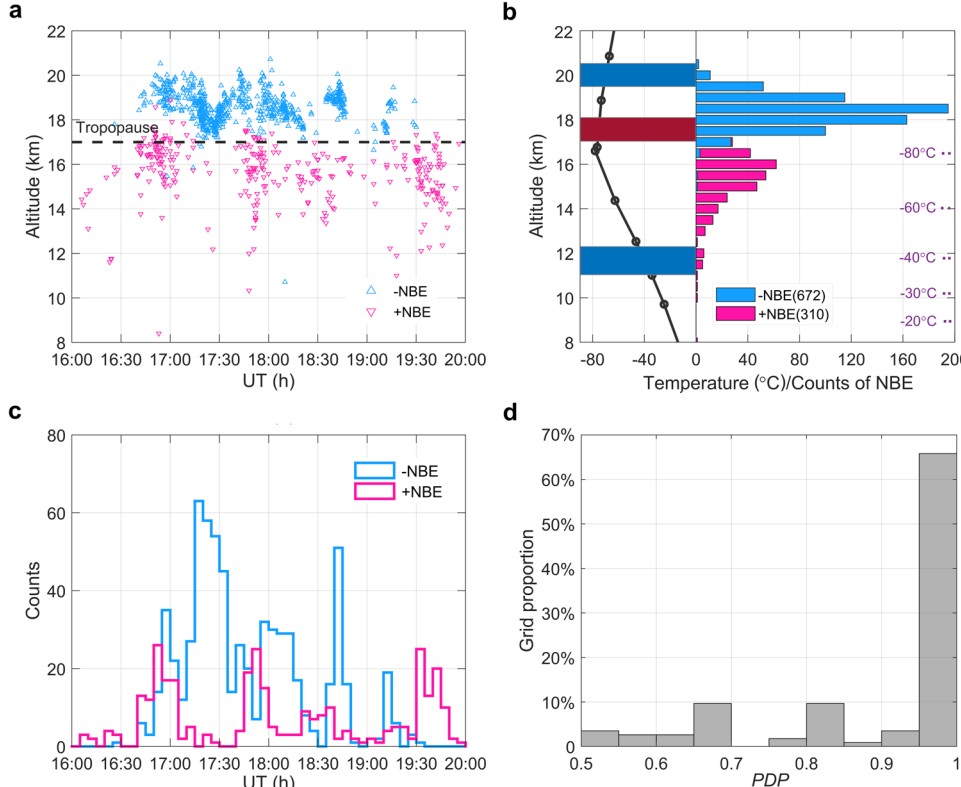

**Fig. 2 | Evolution and distribution of NBEs. a** Height and occurrence variations of NBEs. Blue and red triangles represent negative and positive NBEs, respectively. The identification of NBEs and error evaluation for the NBE height are given in Methods. **b** Histogram for the height of NBEs and the inferred charge region; blue for the negative charge region and red for the positive charge region based on the temperature profile obtained from the balloon sounding profile. **c** Time series of positive NBEs and negative NBEs, respectively; the time bin size is 5 min. **d** Histogram of the proportion of the dominant number of positive or negative polarity to the total number (*PDP*). The grid with the sum of NBEs larger than 5 and at least one negative NBE is counted. A total of 115 grids meet the criteria and are used to count the histogram of the proportion of *PDP*.

ground-based sferics observations). To our knowledge, there is no literature report of the outburst of negative NBEs observed in a tropical storm. As shown in Fig. 2a, the sources of positive NBEs are primarily between 10 km and 17 km and most of the negative NBEs are above the tropopause, suggesting a normally electrified storm with the main negative and positive charge regions, which is consistent with prior NBEs observations[32,39]. The inferred negative charge layer is close to the −40 °C isotherm as measured by the radiosonde from the non-inductive charging mechanism[41,42]. The positive charge region was near the tropopause as suggested by the height distribution of NBEs polarities shown in Fig. 2b, which is higher than generally found in normally electrified thunderstorm[32,43]. The high altitude of the upper positive charge region may contribute to the outburst of negative NBEs[44].

From Fig. 2a, we note that the dominant polarity of NBEs appears to alternate with time. Positive NBEs occur at the beginning of the storm, peaking just before 17:00 UTC where negative NBEs take over. Between 17:10 UTC and 17:40 UTC, positive NBEs occur sporadically whereas negative NBEs occur at a high rate. This alternating pattern continues throughout the sequence. This is further illustrated in Fig. 2c, which presents the number of NBEs, positive and negative in 5-min bins, with time.

We further quantify the relation between positive and negative NBEs by considering cells of $25 \times 25$ km$^2$ and 5-min consecutive time bins. In each space-time cell, we determine the proportion of the dominant number of positive or negative Polarity to the total number (*PDP*), which is defined as the number of NBEs of the dominant polarity to the total number of NBEs[43,45]. A larger *PDP* indicates a larger difference in the proportion of positive and negative NBEs, suggesting greater competition between positive and negative NBEs during a given period (see "Methods", subsection definition of *PDP*). We require a minimum of five events and at least one negative polarity. In all, 115 values are determined. A histogram of *PDP* values is shown in Fig. 2d. More than 80% of the space-time cells have *PDP*-values greater than 0.75, suggesting that one polarity dominates and the strong competition relation of NBEs polarities.

To support that the relation between NBE polarities is general, we have analyzed additional thunderstorms recorded by the Guangzhou (GZ) station for one year in the low-latitude region. On average, the overall occurrence rates of NBEs are about 1.8%. A histogram of *PDP* values is shown in Supplementary Fig. 1. More than 75% of the space-time grids have *PDP* values greater than 0.75, suggesting that the competition behavior of NBEs polarities is general.

**Above-anvil cirrus plumes (AACPs) with negative NBEs**

To understand how the dominant polarity depends on cloud dynamics, we analyze the evolution of the IR brightness temperature provided by the Himawari-8 satellite from 16:50 UTC to 17:50 UTC. This period shows the clearest variations in the dominant polarity and has the least complexity in the evolution of the cloud top temperatures. Fig. 3a shows the cloud top temperature at 17:20 UTC with the NBE activities superimposed. The cloud exhibits a unique temperature shape with a cold ring region encircling a warmer region. This signature suggests that the warm region is an AACP, which is formed by deep convection updrafts lifting clouds into the stratosphere[46–50]. AACP are indicators of severe storms and stratospheric hydration events[49,51]. We see that negative NBEs occur near the warm AACP region where only a few positive NBEs are produced as shown in more detail in Fig. 3b. A radar scan at 17:15 UTC and 17:21 UTC by the weather

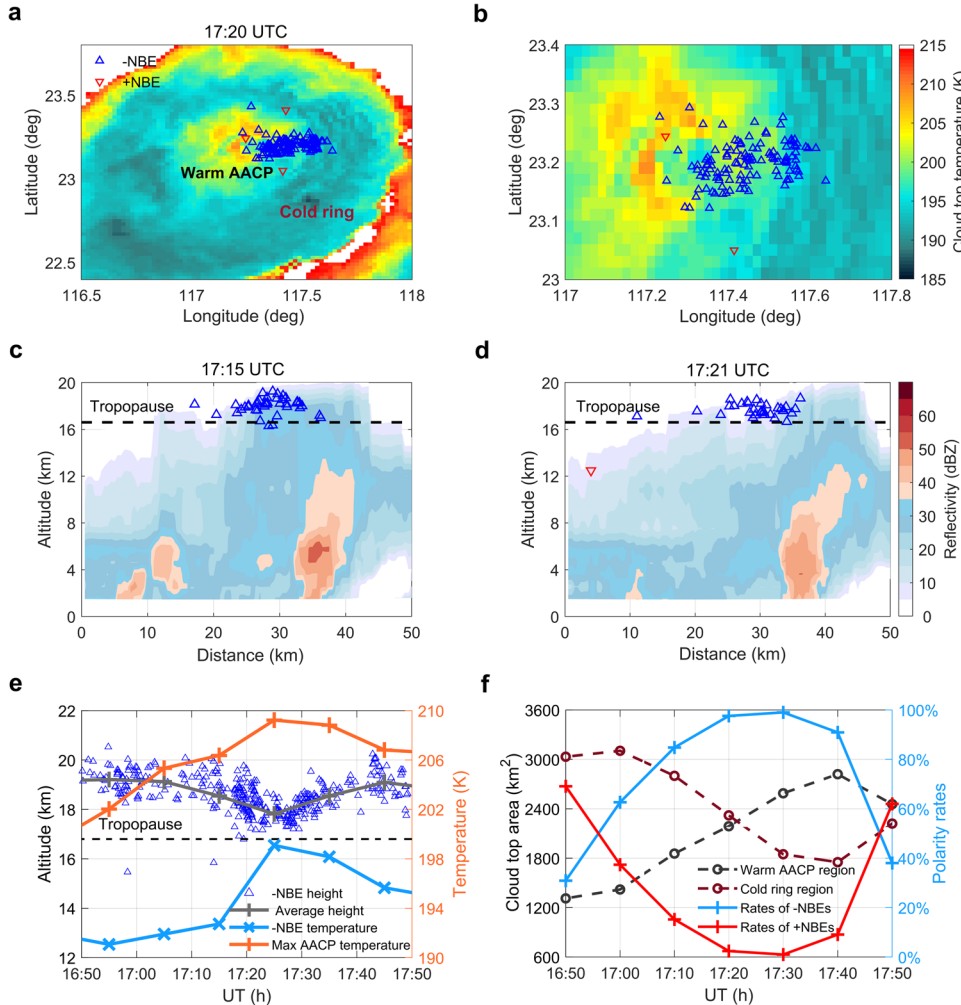

**Fig. 3 | The cloud top condition of NBEs. a** Positions of both polarities of NBEs overlaid on the cloud top temperature at 17:20 UTC. The red '▽' presents positive NBEs while the blue '△' represents negative NBEs. NBEs within 5 min of the time are shown. The warm region bounded by colder IR temperatures is interpreted as AACPs. **b** The same view zoomed in to the NBEs region. **c, d** Vertical cross-section of the radar profile along the outbreak of NBEs at 17:15 UTC and 17:21 UTC, respectively. **e** Comparison between the variations in NBEs height and cloud top temperature. The gray line presents the NBE's height in a 10 min average. The blue and orange lines represent the average cloud top temperature at the NBEs location and the maximum AACP temperature in 10 min, respectively. **f** Variations in cloud top area in different temperature ranges and proportions of NBEs polarity. The area in Supplementary Fig. 2 is counted for calculation.

radar is shown in Fig. 3c, d, with the altitudes of NBEs around the scan time superimposed. The measurements confirm that the clouds reach above the tropopause and that the warm region is an AACP.

For each negative NBE, we determine the cloud top temperature at its location and estimate the maximum AACP temperatures. Fig. 3e compares the 10-min averages of the negative NBE altitudes and temperatures to the AACP temperature. We see that when the AACP temperature increases, the height of negative NBEs decreases and their average temperatures also increase, suggesting that they preferentially occur at the edge of the AACP as it expands.

To investigate this further, we consider a region box of $80 \times 6$ km² around the NBE activity. We label the region within the box with temperatures below 195 K (the tropopause temperature) as the convection cold ring region and the region with higher temperatures as the warm AACP region. The development of the AACP is illustrated in Fig. 3f, which shows the areas of the two regions as functions of time. We note that the rate of negative NBEs increases as the warm AACP region develops, whereas positive NBEs are almost extinguished, suggesting a significant dependence of the dominant polarity on the convective strength and phase of storm cell development. The cloud top temperatures in other periods shown in Fig. 3 are presented in Supplementary Fig. 2.

## Convection phase of the dominant polarity NBEs
To understand how the dominant polarity depends on the convective phase inside the cloud, we analyze the S-band Doppler radar (see "Methods", subsection radar observation). Figure 4a presents the evolution of the maximum height of different reflectivity during the polarity transition from 17:40 UTC to 18:10 UTC. We see that positive NBEs are located relatively deeper in the cloud, and negative NBEs occur in the region with reflectivity 10–15 dBZ near the overshooting top, which is consistent with previous observations[44,52]. The convection stage of overshooting tops (OT) can be separated into two stages: upwelling and decaying[47]. We define the overshooting convective phase as an upwelling stage if the maximum altitude of 40 dBZ gradually increases and finally reaches beyond the tropopause and a decaying stage if the maximum altitude of 40 dBZ decreases and is limited to altitudes more than 3 km below the tropopause[53].

From Fig. 4a, we note that the convective development inside the cloud shows a significant difference for the outburst of negative NBEs than for positive NBEs. The outburst of positive NBEs occurs while negative NBEs are gradually inhibited during the upwelling stage, where the height of 40 dBZ gradually increases reaching an altitude above the tropopause. In particular, as shown in Fig. 4b, negative NBEs surprisingly disappear when the height of 40 dBZ reaches above 17 km,

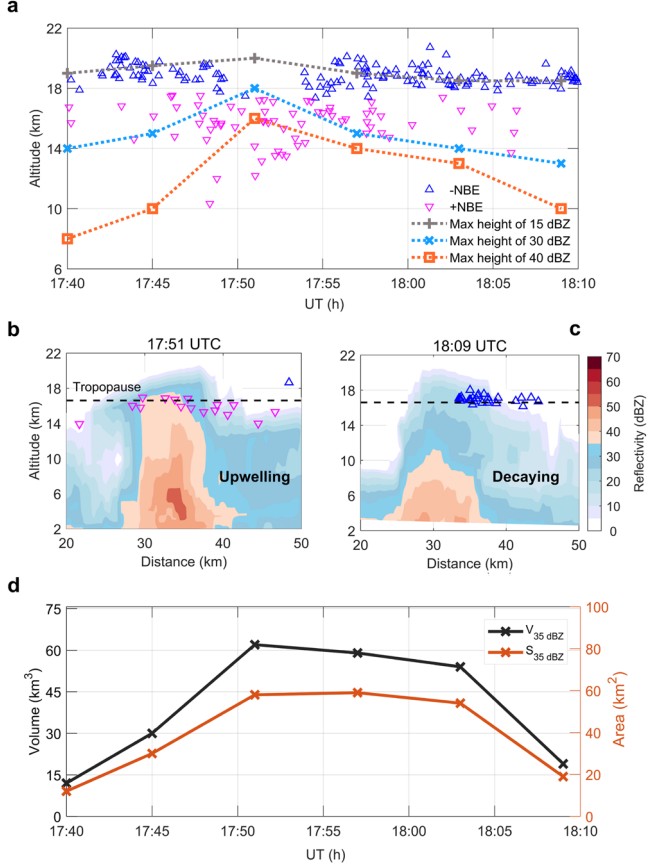

**Fig. 4 | Radar evolution for NBEs polarity transition. a** Evolution of NBEs. The blue '△' presents negative NBEs while the red '▽' represents positive NBEs. The colored dashed line represents the height variations of radar reflectivity. The dashed gray, blue, and orange lines present the maximum height of 15 dBZ, 30 dBZ, and 40 dBZ derived from the vertical cross-section radar, respectively. **b** Vertical cross-section radar profile of NBEs during the upwelling stage. The deep red contour shows a large reflectivity echo (>35 dBZ); the dim blue contour indicates a low reflectivity echo (<35 dBZ). NBEs within 10 km of the cross-section line on the plane position indicator (PPI) is overlaid on the vertical cross-section. The red '▽' presents positive NBEs while the blue '△' represents negative NBEs. **c** Vertical cross-section radar profile of NBEs during the decaying stage. **d** Variations in the area of reflectivity above 35 dBZ ($S_{35dBZ}$) and the volume ($V_{35dBZ}$) of reflectivity above 35 dBZ at altitudes higher than 10 km around NBEs activity.

which is out of the conventional view that negative NBEs are always produced in OT[44,52]. In contrast, negative NBEs outburst while few positive NBEs occur during the decaying stage, where the height of 40 dBZ decreases and generally stays below 13 km as shown in Fig. 4c. The radar in other periods shown in Fig. 4 is given in Supplementary Fig 3. Moreover, we also determine the convection development from the volume of reflectivity above 35 dBZ at altitudes higher than 10 km ($V_{35dBZ}$) and the area of reflectivity above 35 dBZ ($S_{35dBZ}$) around NBEs activity (see "Methods", subsection estimation of the total volume and area of reflectivity). The variations of $S_{35dBZ}$ and $V_{35dBZ}$ as functions of time are illustrated in Fig. 4d. We see that the rate of positive NBEs increases as the $S_{35dBZ}$ and $V_{35dBZ}$ develop, whereas negative NBEs are almost extinguished. This suggests that the variations in altitudes of large reflectivity induced by the convection phase play a major role in the generation of dominant polarity NBE. The radar analysis for other times (see Supplementary Fig. 4) shows similar features.

We further analyze another supercell thunderstorm associated with dominant negative NBEs (see Supplementary Fig. 5), which has been investigated to reveal the connection between negative NBEs and blue discharges[17]. The positive and negative NBEs also exhibit a strong competitive relationship. Negative NBEs occur intermittently when the cloud top reaches above the troposphere and is associated with the low altitude of the large reflectivity (see Supplementary Fig. 6), which is similar to the above tropical storm case.

## A conceptual model for the polarity transition

The aforementioned analysis suggests that the convection strength and phase play a crucial role in the production of the dominant polarity NBEs. As suggested in radar observations, negative NBEs prefer to occur in the decaying stage with a low altitude of large reflectivity while positive NBEs outburst in the upwelling stage with a high altitude of large reflectivity in overshooting thunderclouds. This is consistent with the IR brightness temperature analysis that the rate of negative NBEs increases with the development of the warm AACP region while positive NBEs outburst with the growth of the cold ring region.

We propose a conceptual model that the charge distribution induced by the convective phase is responsible for this result, as illustrated in Fig. 5. In a normally electrified storm, the updraft carries super-cooled droplets and small ice crystals upwards while graupel, which is considerably larger and denser, tends to fall[54]. These different motions of the precipitation would cause collisions to occur. When the rising ice crystals collide with the graupel, the ice crystals become positively charged and the graupel becomes negatively charged, which gradually accumulates a main negative charge layer and an upper positive charge layer[38,42]. Yoshida et al. [55] examined the 3-D lightning location and radar data to investigate the relationship between thunderstorm electrification and storm kinetics. They found that the upper positive charge layer is carried in the updraft with low radar reflectivity while the main negative charge region is located in high reflectivity regions.

Positive NBEs are generally produced between the main negative charge layer and the upper positive charge layer under the stage of the cloud below the tropopause as shown in Fig. 5a. Figure 5b depicts the inferred charge distribution for the outburst of positive NBEs during the upwelling stage of OT. The strong updraft drives the cloud top into the tropopause. Under the intense updraft, the heavier graupels carrying negative charges would be lifted to a higher height as inferred from the high height of large reflectivity in Fig. 4b. Values of reflectivity > 35 dBZ typically indicate the presence of graupel-sized particles[56]. The higher negative charge layer would narrow the gap with the upper positive charge layer, enhancing the ambient electric field. Moreover, the breakdown threshold is lower because of less pressure and humidity for higher altitudes[57], facilitating the initiation of positive NBEs. Meanwhile, ice crystals with the upper positive charge layer are elevated to a very high altitude, which would cause strong mixing with the screening charge layer, making it difficult for the inception of negative NBEs[8].

Figure 5c illustrates the inferred charge distribution for the outburst of negative NBEs during the decaying stage of OT. When the convection gradually decays, the larger and heavier graupels, carrying negative charges, fall into the mixed-phase region, while small ice crystals with positive charges can still be at a high altitude[58], as inferred from the lower altitude of large reflectivity in Fig. 4c. The distance between the upper positive charge layer and the main negative charge layer would then increase, resulting in a lower electric field in this region. This would make it difficult for positive discharge to occur[59]. Meanwhile, the cloud top above the tropopause would accelerate the formation of the screening charge layer[59,60], and the high upper positive charge and the enhancement of the electric field by the accumulated positive charge make it easier for negative NBEs to win the competition for the upper positive charge layer. Moreover, the reduced intracloud discharge may lead to further accumulation of the main negative charge layer, which would make it easier to produce more negative CGs. This may explain why frequent negative CGs were observed during blue discharges[17,19].

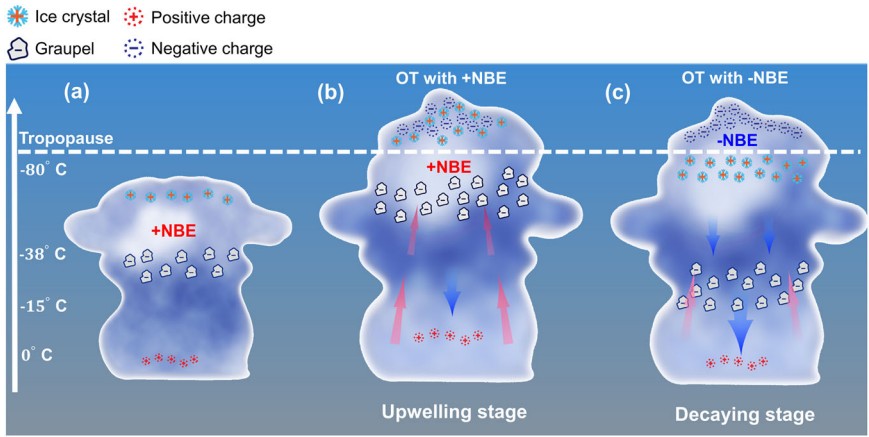

**Fig. 5 | Conceptual model for the illustration of NBEs polarity transition in thunderclouds with OT. a** A normally electrified thundercloud with the upper positive charge of ice crystals and the main negative charge of graupels. **b** The required charge distribution for the dominant positive NBEs during the upwelling stage with strong convection of OT. Positive NBEs outbreak inside the cloud between the middle negative charge layer and the upper positive charge layer. The size of the red arrow represents the strength of the updraft while the blue arrow represents the downdraft. **c** The required charge condition for the outbreak of negative NBEs during the decaying stage of OT. Negative NBE discharge initiates in the upper region of the cloud between the upper positive and negative screen charge layer.

The conceptual model is based on the well-established simplified charge layer distribution of the normally electrified thunderstorm, which is inferred from radar and NBE distributions based on the relation between the storm electrification and kinetics[42,54,55]. It should be noted that the conceptual model has some limits as the charge structure in some thunderclouds is more complex because of the highly strong wind shear inside the cloud[56], which may result in forming small charge pockets and initiating sporadic positive NBEs with higher heights than negative polarity. This complexity, though, is outside of the scope of this paper and will require further investigation. Nonetheless, the conceptual model provides a reasonable and unified illustration for the finding of NBE's polarity competition in overshooting thunderclouds.

## Discussion

Krehbiel et al. [8] proposed a mechanism based on quasi-electrostatic fields to demonstrate how cloud top discharges initiate in the thundercloud. They suggested that blue discharges (referring to blue starters and jets) occur as a result of an electrical breakdown between the upper positive charge and the screening charge after a CG or IC discharge inducing a sudden charge imbalance. If the upper positive charge layer is strongly mixed into the screening charge. The situation produces a substantial imbalance between the main negative and positive charges favorable for the occurrence of gigantic jets[59,61].

Our results provide insight into the mechanism of cloud top discharges from the view of the dynamic development of convection. We emphasize the importance of the convection phase on the initiation of cloud top discharges. The competition of NBE polarities in our findings suggests that the height variations of the upper positive charge layer induced by the strength of updrafts would play a significant role in the initiation of types of cloud top discharges. Negative NBEs occur as the onset of blue discharges[17–19], and positive NBEs serve as the initial stages of gigantic jets[62–64]. The upwelling stage of the overshooting cloud with an outburst of positive NBEs in Fig. 5b is associated with a very high altitude of the upper positive charge layer causing strong mixing with the screening layer, which potentially triggers negative gigantic jets occurring beneath the upper positive charge region[59]. The gigantic-jet condition is indeed usually featured with 40 dBZ echoes above 14 km[65,66], which is consistent with the proposed conceptual model. In contrast, the decaying stage of the OT with an outburst of negative NBEs in Fig. 5c is associated with a relatively stable charge layer at the cloud top, which would facilitate the production of blue discharges. Recent measurements from space indeed suggest that blue discharges are produced in overshooting clouds with a low altitude of high reflectivity[4,19]. These findings would provide the constraints on the generation mechanism of cloud top discharges.

We highlight the finding that the occurrence of AACPs is correlated well with the lightning jump of negative NBEs at the cloud top. Cirrus plumes are generated under strong convection lifting clouds into the stratosphere and forming cirrus clouds via the gravity wave-breaking process[49,50] or hydraulic jump[51]. They play a major role in greenhouse gas injection into the stratosphere, especially for water vapor with increases leading to a cooling of the stratosphere and warming of Earth's surface[67,68]. Recent model estimates of the water vapor transport deep into the stratosphere for one jump of cirrus plumes may exceed 7 tons/s[51]. The close relationship between NBEs and AACPs in our findings suggests that ground-based observations of radio signals would provide a new means to measure the impacts on the composition of the upper troposphere and lower stratosphere. Positive NBE outbursts could provide an early precursor for cirrus plume production as they usually occur during the upwelling stage of the deep convection lifting cloud into the stratosphere. Negative NBEs can be a good proxy for the occurrence of cirrus plumes as well as a good estimate of the masking anvil cloud top height.

Additionally, cloud top discharges can directly affect the exchange of greenhouse gases between the troposphere and stratosphere through the production of nitrous oxide, and the depletion of stratospheric ozone[13,15]. The production of kilometer-scale blue discharges would lead to 1660 moles per discharge[2]. However, this region of the atmosphere is difficult to access experimentally. Climate models suffer from large uncertainties in estimating global perturbations of greenhouse gases in the stratosphere due to unrealistic parameterization of cloud top discharges[12]. Our conceptual model provides a unified illustration of the required convection conditions for cloud top discharge generation. The implication is that we may, in the future, develop proxies for their generation rate in global storms measured by the new generation meteorological satellites to predict the regional and global effects they may have on the lower stratosphere[14]. From that perspective, this offers the possibility to evaluate the global impact of cloud top discharge on the greenhouse gas agents at the tropopause.

## Methods

### Ground-based sferics observations

The VLF/LF sferic signal is measured by receivers of the Jianghuai Area Sferic Array (JASA). Each station is equipped with a vertical $E$-field antenna (with 3 dB bandwidth of 800–300 kHz) and records $dE_z/dt$ (time derivative of the vertical electric field) for each lightning event. All recordings are synchronized with a GPS clock with a sampling rate of 5 MHz. The atmospheric electricity sign convention is used for the $E$-field sensor; that is, positive NBEs occurring between the upper positive charge layer and main negative charge layer produce the negative initial peak while negative NBEs occurring between the upper positive charge layer and screening charge layer produce the positive initial peak.

### Identification and detection efficiency of NBEs

NBEs are characterized by a shorter duration (<40 μs) of the VLF/LF signal compared to normal lightning. They are always associated with ionospheric reflected waves at distant stations, especially during the nighttime. We adopt the criterion of the short duration of the pulse (40 μs), the reflected wave in distant stations, and the high signal-to-noise ratio to identify NBEs. Each selected event is manually inspected to avoid misidentification.

Data from a local lightning network consisting of 19 Vaisala LS7000 sensors, the GLD360 network, and JASA are combined to determine each NBE location. We first use the identified NBEs to match the location provided by GLD360 and the local lightning array. Nearly 80% of NBEs in the tropical storm recorded by VLF/LF stations can be matched with the local network and GLD360. If missed, we then use at least four VLF/LF stations to locate the event. Although the VLF/LF stations are located 400–1000 km away from the tropical storm, which would cause considerable attenuation of direct ground waves of NBEs signal, the signal can also be triggered by obvious ionospheric sky-waves (shown in Fig. 1c). The location error for the tropical storm in this paper is within 5 km. To evaluate the detection efficiency for the observed NBEs, we present the peak current distribution of the matched NBEs shown in Supplementary Fig. 7.

### Error evaluation for the NBE height

The uncertainty of the estimated NBE heights mainly originated from the error in the source location and the reflected pulses. The sampling rate of the radio receiver is 5 MHz/s (0.2 μs point-to-point), and the identification of the pulses of the reflected signal is manually checked to avoid misidentification. To estimate the error of the NBE height, we performed the following test. For a typical NBE, up to 5 km and 1-μs are intentionally added in 2-D location and reflection pulses. Under such conditions, the results suggest that the maximum error in altitude is no larger than 0.8 km variance from the original altitudes (see Supplementary Fig. 8). It should be noted that the height of NBEs is derived by assuming that it is a point discharge, regardless of the channel length.

### IR brightness and radar observation

The IR brightness data are obtained from the geostationary meteorological satellite Himawari-8, which has ten IR channel bands ranging from 3.9 μm to 13.3 μm. The window-IR (IR; 10.4 μm) channel is used to sense the cloud top temperature with a temporal resolution of 10 min.

Radar observations are derived from an S-band radar. The beamwidth of the radar is ≤1°. The detection height includes nine layers of elevation angles ranging from a minimum of 0.5° to a maximum of 19.5° and a volume scan on a 6-min cycle. The distance between the radar and the storm in most of the life cycle varies from 80 km to 120 km, corresponding to measuring the vertical height from 2 km to 22 km. The detection precision for the horizontal reflectivity ($Z_H$) is 1 dB. Radar coordinate data are processed by spline interpolation into a Cartesian grid with a horizontal resolution of 0.25 × 0.25 km and a vertical resolution of 0.25 km.

### Definition of PDP

To quantify the relation between positive and negative NBEs, we define the *PDP*:

$$PDP = \frac{\max(+\text{NBE}, -\text{NBE})}{\sum(+\text{NBE}, -\text{NBE})} \qquad (1)$$

A larger *PDP* indicates a larger difference in the proportion of positive and negative NBEs, suggesting greater competition between positive and negative NBEs during a given period. For example: 10 NBEs, including 8 positive and 2 negative NBEs, are generated in a period. The value of *PDP* in this period is $PDP = \frac{\max(8,2)}{\sum(8,2)} = \frac{8}{10} = 0.8$. Note that we only calculate the value of *PDP* during the period when the sum of NBEs is larger than five and at least one negative NBE.

### Estimation of the total volume and area of reflectivity

According to the size of the storm cell producing NBE activity, we restrict a region box of 30 × 40 km² around the NBE activities. We first establish a spatial Cartesian coordinate system in the box and project the radar volume scan data onto the coordinate. Then, the spatial area with an altitude between 10 km and 22 km is divided into cubic grids with a resolution of 0.5 km. The nearest neighbor interpolation method is applied to obtain the radar reflectivity of the center point of each grid. Finally, we count the number of grids with radar reflectivity greater than 35 dBZ to calculate the value of $V_{35\text{dBZ}}$ and put these grids onto a horizontal plane to obtain the value of $S_{35\text{dBZ}}$.

## Data availability

The datasets used in this study are freely available in the open-access depository (https://doi.org/10.5281/zenodo.13222010). All data supporting the findings of this study are available within the paper and are also provided as a Source Data file. Source data are provided in this paper.

## Code availability

The codes that support the findings of this study are fully available from the author (feifan@ustc.edu.cn) upon request.

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

## Acknowledgements

The authors would like to acknowledge financial support from the National Basic Science Center of the National Natural Science Foundation of China (42188101), the Project of Stable Support for Youth Team in Basic Research Field CAS (YSBR-018), the National Natural Science Foundation of China (42005068 and 42375074), and the Basic Research Fund of CAMS (2023Z008). ASIM team is funded by ESA and by national grants of Denmark. C.K. is funded by the Independent Research Fund Denmark (grant 1054-00104). We thank Vaisala for the GLD360 data. We thank Ryan Said for his helpful discussion on GLD360 data.

## Author contributions

F. Liu processed the data, produced the figures, and drafted the manuscript. T.N., O.C., G.L., T.W., F. Lyu, C.K., D.L., B.Z., and J.L. revised the manuscript. F. Liu, G.L., F. Lyu, B.Z., and W.L. conducted the ground-based electric field measurements. F. Lyu and W.L. provided the radar and Vaisala data. All authors discussed and supported the conclusions.

## Competing interests

The authors declare no competing interests.
