## [Peer Review File · Nature Communications]

Editorial Note: Parts of this peer review file have been redacted as indicated to avoid any copy right infringement.REVIEWER COMMENTS

Reviewer #1 (Remarks to the Author):

Review of the article titled "Altitude and polarity transitions of corona activity in thundercloud tops reaching the lower stratosphere" submitted for publication in Nature Communications by Liu et al.

The article under consideration presents a meteorological analysis of a storm producing copious amounts of Narrow Bipolar Events (NBEs). This reviewer does not find this article suitable for publication in Nature Communications due to the following reasons:

1. The findings lack novelty and potential impact. The key findings regarding the fact that positive and negative NBEs have a bi-modal altitude distribution, facilitated by distinct storm charge structures, have already been discussed in the peer-reviewed literature. The NBE polarity convention used has to do with the sign of field change.
2. On the one hand, the mechanism to explain the shift in the dominant NBE polarity due to the evolution of convection during the storm lifetime seems obvious and, thus, quite reasonable. On the other hand, it does not explain the observed temporal dynamics unambiguously. The conclusions are largely dependent on indirect measurements of the storm's electrical structure.
3. The conclusions are based on the analysis of a single storm, and there is no evidence to suggest that the conceptual model should be general. The only other example provided in the Supporting Information does not seem to corroborate the conclusions. Additionally, the Extended Data Figure 1 suggests that the model should not be one-dimensional, i.e., that there are two horizontally-separated regions where NBEs of different polarities are produced.
4. The manuscript lacks a contemporary understanding of NBE physics. NBEs are produced by large and fast streamer systems of both polarities. Particularly, even a single NBE polarity, let's say positive NBEs, can be produced by streamer discharges of both polarities.
5. All potential connections between NBEs and jets mentioned in this manuscript have been previously discussed in the peer-reviewed literature. Including in papers overlooked in the authors' reference list.

Reviewer #2 (Remarks to the Author):

Review of the manuscript entitled "Altitude and polarity transitions of corona activity in thundercloud tops reaching the lower stratosphere" by Liu et al.

The authors present a detailed analysis of a thunderstorm producing enormous amount of narrow bipolar events (NBE). The authors show the changing prevalence of one polarity and place it in the changing properties of the storm cells evolution.

The manuscript is well written, presents new observations, and a comprehensive analysis. The authors effectively combine electromagnetic measurements with radar observations, the cloud properties are derived from the satellite measurements.

The results are original and worth to publish, the work is significant in the field of atmospheric electricity and related meteorology. The methodology is quite well described. There is enough details the authors provided to support their statements.

I think that the manuscript needs some improvements and clarification before it can be considered for publication in the Nature Communications journal. My comments are listed below.

1. Title: From the title (and also when looking at the names of co-authors from the ASIM group) I would expect the manuscript contains also the optical data confirming the existence of corona by observation of blue discharges. As this is not the case, and the presence of corona is based on simultaneous previous observations of NBEs and blue discharges, I would ask the authors to consider the modification of the title accordingly.

2. You state that "high lightning activity measured by the GLD360 network, and the domination of negative cloud to-ground (-CG) and positive intra-cloud (+IC) discharges..." but you did not present anything about these discharges. It would be interesting to see what was their distribution in time and space in comparison with the NBEs of both polarities.

3. You discuss the changes in the charge arrangement leading to the NBE polarity switching. It would be useful to formulate clearly what generally caused the observed enormous amount of NBEs and how often this can happen, especially if you want to use the presence of radio signals emitted by NBEs as an indicator of an exchange of greenhouse gases between the troposphere and stratosphere.

4. Methods: The estimation of the altitude of NBE is a crucial part of the work. I would suggest to evaluate more precisely the error in the altitude calculation."

5. Fig.4a: I do not understand why the NBEs appeared only in groups which are separated by approx. 1 min of nothing, this cannot be natural, please explain.

6. I is quite confusing that you use in different figures different time intervals, if there is a reason for such approach, could you explain it properly?

Reviewer #3 (Remarks to the Author):

This is a well-written paper. I have a few comments that the authors need to consider and address in the revised version:

1. Lines 83-83: Kindly reword this sentence, I recommend inserting: "helps estimate" after "...and the reflected ones".

2. Lines 85-86 and Figure 2a. It appears that a significant number of the negative NBEs are above the troposphere. The authors should note here that previous observations have also reported above-tropopause NBEs in "convective surges (plumes) overshooting the tropopause" such as Smith et al. (2004) doi:10.1029/2002RS002790 and Nag et al. (2010) doi:10.1029/2009JD012957.

3. Line 107: The acronym PDP and its meaning is not introduced before in the text.

4. Lines 109-111: It is difficult to understand the significance of this finding without further explanation of what PDP means and how values greater than 0.75 is suggestive of "strong competition".

5. Lines 196-197: Need to discuss the reasoning for this statement regarding the lower breakdown threshold. Is it because of changes in atmospheric composition, pressure, and temperature? Perhaps need a reference like Rioussset et al. (2020) doi.org/10.1016/j.icarus.2019.113506?

6. Line 210-213: What about the occurrence of "regular" ICs between the main negative and positive charge regions. They would also play a role (perhaps a more dominant one) than the negative CGs in this charge balance? Please clarify here.

Point-by-point Response to Reviewer's Comment

Manuscript number: NCOMMS-23-46957-T

Manuscript Title: Altitude and polarity transitions of corona activity in thundercloud tops reaching the lower stratosphere

Reviewer #1:

Review of the article titled "Altitude and polarity transitions of corona activity in thundercloud tops reaching the lower stratosphere" submitted for publication in Nature Communications by Liu et al.

The article under consideration presents a meteorological analysis of a storm producing copious amounts of Narrow Bipolar Events (NBEs). This reviewer does not find this article suitable for publication in Nature Communications due to the following reasons:

We appreciate the referee's insight and thoughtful comments. Although the other two referees provided a very positive evaluation on this work and suggested that it could potentially have a high impact on the geophysical research community, we now further emphasize the novelty of this work in the revised version to clarify and address each of the comments and concerns you have raised. Below is our point-by-point response to each comment.

1. The findings lack novelty and potential impact. The key findings regarding the fact that positive and negative NBEs have a bi-modal altitude distribution, facilitated by distinct storm charge structures, have already been discussed in the peer-reviewed literature. The NBE polarity convention used has to do with the sign of field change.

Response: We agree with the referee's point that several studies have reported the altitude distribution of NBEs, and those papers are appropriately cited in the introduction. It should be noted that prior observations, mainly conducted with respect to thunderstorms at middle and high latitude, generally focused on positive NBEs in the cloud (Lv et al., 2010; Nag et al., 2010; Wu et al., 2011, 2012; Karunarathne et al., 2015; Bandara et al., 2023; Li et al., 2023). A latest study by Chen et al., (2024) present five clusters of upper-level compact intracloud discharges (CIDs) in overshooting tops. Based on the data analysis and theoretical work, they infer that upper-level CIDs were likely made possible by the recurring action of energetic cosmic rays and the rapid recovery of the negative screening charge layer. However, what meteorological and electrical conditions of thunderstorms produce the outbreak of negative NBEs on cloud top and the evolution of both polarities at such condition are still unclear.

To our knowledge, there has been no literature ever reported NBEs polarities relationship. In our work, we primarily focus on the evolution of NBEs polarities in a tropical storm with an extraordinarily high number of negative NBEs. It is the first

study that shows the dominant polarity changes with time during a storm. The novelty and broad impact are summarized as follows:

The novelty:

- 1. It is the first time we observe an unprecedentedly large number of negative corona discharges in a tropical storm cloud top penetrating the stratosphere.*
- 2. We first show the dominant NBEs polarity changes with time during a storm, and present the strong competition relationship between NBEs polarities in overshooting thundercloud, as well as their polarity generation depending on the storm cell development phase is reported.*
- 3. The lightning jump of negative NBEs is found to be closely associated with the occurrence of above-anvil cirrus plumes (AACP) in the lower stratosphere.*

The broad impact:

- a) Our results provide new insight into the generation of cloud top discharges from the view of the dynamic development of convection. Theoretical work based on the quasi-electrostatic field simulation predicts that the generation of cloud top discharges depends on a sudden charge imbalance by normal lightning. The competition of NBEs polarities in our finding suggests that the strength degree of updraft would play a significant role in the initiation of cloud top discharges, which provide constraints on their generation mechanism.*
- b) The proposed conceptual model provides a unified illustration of the required convection phase for cloud top discharge generation. The implication is that we may, in the future, develop proxies for their generation rate in global storms measured by the new generation meteorological satellites to predict the regional and global effects they have on the lower stratosphere. From that perspective, this offers the possibility to evaluate the global impact of cloud top discharge on the greenhouse gas agents at the tropopause.*
- c) The close relationship between the occurrence of AACP and the outbreak of NBEs in our findings suggests that ground-based observations of radio signals would provide a new means to measure the impacts on the concentrations of greenhouse gases in the upper troposphere and lower stratosphere.*

Inspired by the reviewer's comments, we realized that we did not clearly present the innovation. We have now changed the title and revised the text to emphasize the novelty and broad impact.

Please see revised manuscript (Lines 1-2, Lines 242-246, and Lines 278-284):

Change of title: Polarity transitions of narrow bipolar events in thundercloud tops reaching the lower stratosphere.

Our results provide new insight into the mechanism of cloud top discharges from the view of dynamic development of convection. We emphasize the importance of convection phase on the initiation of cloud top discharges. The competition of NBEs polarities in our findings suggests that the height variations of the upper positive charge layer induced by the strength of updrafts would play a significant role in the initiation of cloud top discharge types.

The conceptual model provides a unified illustration of the required convection phase for cloud top discharge generation. The implication is that we may, in the future, develop proxies for their generation rate in global storms measured by the new generation meteorological satellites to predict the regional and global effects they may have on the lower stratosphere. From that perspective, this offers the possibility to evaluate the global impact of cloud top discharge on the greenhouse gas agents at the tropopause.

Add the reference: Chen, S., Rakov, V. A., Zhu, Y., & Ding, Z. Clusters of compact intracloud discharges (CIDs) in overshooting convective surges. *J. Geophys. Res. Atmos.*, 129, e2023JD040307 (2024).

2. On the one hand, the mechanism to explain the shift in the dominant NBE polarity due to the evolution of convection during the storm lifetime seems obvious and, thus, quite reasonable. On the other hand, it does not explain the observed temporal dynamics unambiguously. The conclusions are largely dependent on indirect measurements of the storm's electrical structure.

Response: Thanks for the support of the proposed conceptual model. We agree with the reviewer's point that the electrical structure in the conceptual model is inferred from the indirect measurements of radar and NBEs distributions. It should be noted that many observation and simulation results have suggested that thunderstorm electrification is related to storm kinetics parameters such as graupel volume and updraft from the radar observations (Carey and Rutledge, 1996; Bruning et al., 2010; Calhoun et al., 2013, 2014; MacGorman et al., 2017). Yoshida et al., (2017) examined 3-D lightning location BOLT data and radar data to investigate the relationship between thunderstorm electrification and storm kinetics. As shown in Figure R1, they found that BOLT sources ascribed to the upper positive charge region are located in the radar reflectivity region less than 30 dBZ. These observations imply that the upper positive charge layer is carried in the updraft with low radar reflectivity. In contrast, BOLT sources ascribed to the main negative charge region are located in high reflectivity regions.

As such, based on the relation between storm electrification and kinetics, the electrical structure of our conceptual model can be inferred from the NBEs distributions and radar observations. We agree that there are some limits of the model due to such indirect measurements. Nevertheless, this provides a simplified and - as the referee has pointed out - reasonable illustration on the finding of competition relation between NBEs polarities in overshooting tops.

[redacted]

The conceptual model can explain the observed temporal dynamics of NBEs polarities

As shown in the radar-inferred conceptual model on the left of Figure R2, the strong updraft driving the heavier graupels with negative charges would be lifted to a higher height. This would narrow the gap with the upper positive charge layer, enhancing the ambient electric field and making it easier to initiate positive NBEs. Meanwhile, ice crystals with the upper positive charge layer are elevated to a very high altitude, which would cause strong mixing with the upper screening charge layer, making it difficult for the inception of negative NBEs. This can explain the occurrence of the outbreak of positive polarity but few negative NBEs.

When the convection gradually decays as shown on the right of Figure R2, the larger and heavier graupels, carrying negative charges, fall into the mixed-phase region, while small ice crystals with positive charges can still be at a high altitude. The distance between the upper positive charge layer and the main negative charge layer would then increase, resulting in a lower electric field in this region. This would make it difficult for positive discharge to occur. Meanwhile, the high upper positive charge and the enhancement of the electric field by the accumulated positive charge make it easier for negative NBEs to win the competition for the upper positive charge layer. This can explain the occurrence of the outbreak of negative polarity but few positive NBEs.

Figure R2. The conceptual model for illustration of competition of NBEs polarities in overshooting thundercloud tops

Inspired by the reviewer’s comments, we realized the possible limits of the proposed model from the radar indirect measurements. We have now added the discussion. It merits further investigation on the relation between storm kinetics and electrical structure of storm producing outbreak negative NBEs based on the direct 3-D lightning and high temporal resolution radar data.

Please see revised manuscript (Lines 196-199 and Lines 225-233):

Yoshida et al., (2017) examined 3-D lightning location data and radar data to investigate the relationship between thunderstorm electrification and storm kinetics. They found that the upper positive charge layer is carried in the updraft with low radar reflectivity while the main negative charge region is located in high reflectivity regions.

The conceptual model is based on the well-established simplified charge layer distribution of the normally electrified thunderstorm, which is inferred from the radar and NBEs distributions based on the relation between storm electrification and kinetics. It should be noted that the conceptual model has some limits as the charge structure in some thunderclouds is more complex, which may result in forming small charge pockets and initiating sporadic positive NBEs with higher heights than negative polarity. This complexity, though, is outside of the scope of this paper and will require further investigation.

3. The conclusions are based on the analysis of a single storm, and there is no evidence to suggest that the conceptual model should be general. The only other example provided in the Supporting Information does not seem to corroborate the conclusions. Additionally, the Extended Data Figure 1 suggests that the model should not be one-dimensional, i.e., that there are two horizontally-separated regions where NBEs of different polarities are produced.

Response: Thanks for the comment. The thunderstorm producing outbreak of negative NBEs is relatively rare due to the strict conditions for negative NBEs generation. To our knowledge, there has not been any literature ever reported such a large number of negative NBEs in a tropical storm top, which makes this study extremely novel and timely.

To clearly show the consistency with the other storm, we now give a detailed description of the other storm shown in Figure R3. First, the storm is dominated by negative NBEs, which can be used to investigate the relation of NBEs polarities. Figure R3c shows that the different NBE polarities occur alternatively during the storm evolution, also suggesting the strong competition of the polarity. Secondly, the radar analysis shown in Figure R3a suggests that negative NBEs outbreak under the condition with a low altitude of 40 dBZ, which is consistent with the proposed conceptual model.

Figure R3. Detailed information of the other thunderstorm with outbreak of negative NBEs.

To support that the competition relation is general, we statistically analyze more thunderstorms recorded by the GZ station for one-year in the low-latitude region. By considering cells of $25 \times 25 \text{ km}^2$ and 5-minute consecutive time bin, we determine the Proportion of the Dominant number of positive or negative Polarity to the total number (PDP) in a grid required with a minimum of 5 NBEs and both polarities. In all, 1574 grids are determined. A histogram of PDP values is shown in Figure R4. Approximately 80 % of the space-time grids have PDP-value greater than 0.7, suggesting that one polarity always dominates and the strong competition relation of polarities is general, which thus supports the results of the manuscript.

Figure R4. Histogram of the proportion of competition value PDP in the grids.

Please see revised manuscript (Lines 115-119):

To support that the relation between NBE polarities is general, we have analyzed additional thunderstorms recorded by GZ station for one-year in the low-latitude region. On average, the overall occurrence rate of NBEs is about 1.8%. A histogram of PDP values is shown in Extended Data Fig. 1. More than 75 % of the space-time grids have PDP-value greater than 0.75, suggesting that the competition relation of NBEs polarities is general.

4. The manuscript lacks a contemporary understanding of NBE physics. NBEs are produced by large and fast streamer systems of both polarities. Particularly, even a single NBE polarity, let's say positive NBEs, can be produced by streamer discharges of both polarities.

Response: Thanks for the comment. Understanding the physical mechanism of NBEs requires high spatial and temporal radio measurements from interferometer and lightning mapping imaging. Rison et al., (2016) used a broadband RF interferometer to investigate NBEs and lightning initiation in New Mexico thunderstorms. They found that NBEs are produced by fast positive breakdown (FPB) that propagates away from thundercloud positive charge and toward thundercloud negative charge at a speed of $\sim 5 \times 10^7$ m/s. A later study of Florida thunderstorms by Tilles et al. (2019) suggests that NBEs can also be caused by the fast negative breakdown (FNB) that propagates instead away from the negative charge and toward the positive charge at a similar speed. Further, Lyu et al., (2019) suggest that lightning initiation can also begin by a process with isolated short VHF pulses of typical duration less than 0.5 μ s.

For the physical mechanism of NBEs why they can also produce strong HF/VHF radiation. Liu et al., (2019) first developed a statistical approach to investigate the radio spectrum of NBEs, and suggested that an ensemble of tens of millions of streamers can

reproduce the current moment, charge transfer, and radio spectrum of fast breakdown. The recent optical measurements by ASIM from ISS confirm their result. Liu et al., (2021) presented both polarities of NBEs associated with the 337 nm but without or weak 777.4 nm, suggesting the streamer breakdown.

Our work focuses on the phenomenal features of NBEs polarities and investigates the relationship between NBE polarity and their dynamic meteorology. NBEs can serve as the initiation of normal lightning and cloud top discharges in deep convection penetrating the lower stratosphere, understanding what conditions of thunderstorms can produce NBEs, especially negative polarity has implications not only for understanding lightning initiation but also for studying their perturbations to the concentrations of greenhouse gases in the upper troposphere.

We observed an unprecedentedly large number of negative NBEs in a tropical storm cloud top penetrating the stratosphere, and find the competitive relationship between the occurrence of NBEs of different polarities. The comprehensive electrical and meteorological analysis suggests that the positive charge layers dominated by convection strength contribute to NBEs polarity generation. This would provide the constraints on the generation mechanism of cloud top discharges.

We added the contribution of our results to understanding physical insight in the discussion.

5. All potential connections between NBEs and jets mentioned in this manuscript have been previously discussed in the peer-reviewed literature. Including in papers overlooked in the authors' reference list.

Response: We agree with the referee that the connections between NBEs and jets have been discussed in the literature. In this work, we focus on the competition between negative and positive NBEs rather than NBEs and jets. The connection between NBEs and jets mentioned in our work is used to discuss what conditions produce blue discharges and gigantic jets as well as their difference.

According to the theoretical prediction from Krehbiel et al. (2008) and Rioussset et al., (2010), blue discharges occur as a result of an electrical breakdown between the upper positive charge and the screening charge after a CG or IC discharge inducing a sudden charge imbalance. If the upper positive charge layer is strongly mixed into the screening charge, such situation would produce a substantial imbalance between the main negative and positive charges favorable for the occurrence of gigantic jets.

The competition of NBEs polarities in our findings suggests that the height variations of the upper positive charge layer induced by the strength of updrafts would play a significant role in the initiation of types of cloud top discharges. The upwelling stage of the overshooting cloud with the outburst of positive NBEs is associated with a very high altitude of the upper positive charge layer causing strong mixing with the screening

layer, which potentially triggers negative gigantic jets occurring beneath the upper positive charge region. The gigantic-jet condition by Boggs et al., (2018) is usually featured with 40 dBZ echoes above 14 km, which is consistent with our proposed model. In contrast, the decaying stage of the overshooting tops with outburst of negative NBEs in Fig. 5 (right) is associated with a relatively stable charge layer at the cloud top, which would facilitate the production of blue discharges.

These results may provide new insight into the mechanism of cloud top discharges from the view of the dynamic development of convection, with implications for studies of the chemical perturbations of greenhouse gas concentrations by cloud top discharges at the tropopause.

Reviewer #2:

Review of manuscript entitled “Altitude and polarity transitions of corona activity in thundercloud tops reaching the lower stratosphere” by Liu et al.

The authors present a detailed analysis of a thunderstorm producing enormous amount of narrow bipolar events (NBE). The authors show the changing prevalence of one polarity and place it in the changing properties of the storm cells evolution.

The manuscript is well written, presents new observations, and a comprehensive analysis. The authors effectively combine electromagnetic measurements with radar observations, the cloud properties are derived from the satellite measurements.

The results are original and worth to publish, the work is significant in the field of atmospheric electricity and related meteorology. The methodology is quite well described. There is enough details the authors provided to support their statements.

I think that the manuscript needs some improvements and clarification before it can be considered for publication in the Nature Communications journal. My comments are listed below.

Response: Thanks for your support and constructive comments, which have greatly helped to improve this paper. We have revised the title and analysis sections, taking into consideration your feedback. The detailed point responses are listed in below.

1. Title: From the title (and also when looking at the names of co-authors from the ASIM group) I would expect the manuscript contains also the optical data confirming the existence of corona by observation of blue discharges. As this is not the case, and the presence of corona is based on simultaneous previous observations of NBEs and blue discharges, I would ask the authors to consider the modification of the title accordingly.

Response: We agree with the referee’s comment. the term NBEs is usually defined from radio signal. Owing to the ISUAL and ASIM optical measurements, we now know that the source is a corona discharge in the cloud producing both NBEs in radio signal and blues in optical. We have used ‘corona discharge’ in the title based on the close relation between NBEs and corona discharges reported by several literatures from a broader impact.

As there are no simultaneous optical observations, we now modify the title to “Polarity transitions of narrow bipolar events in thundercloud tops reaching the lower stratosphere” according to the referee’s suggestion.

Please see revised title (Lines 1-2):

Polarity transitions of narrow bipolar events in thundercloud tops reaching the lower stratosphere.

2. You state that “high lightning activity measured by the GLD360 network, and the domination of negative cloud to-ground (-CG) and positive intra-cloud (+IC)

discharges...” but you did not present anything about these discharges. It would be interesting to see what was their distribution in time and space in comparison with the NBEs of both polarities.

Response: The referee brings up an excellent point. Following the referee’s suggestion, we have analyzed the distributions of CGs and ICs with different polarities of NBEs within a 25×25 km grid in a given 5-min bin. A total of 114 grids with negative NBEs and 104 grids with positive NBEs are determined. The results are shown in Figure R5, it is interesting to note that negative NBEs are more correlated with -CGs and +ICs compared to positive NBEs.

Figure R5. Relationship among -CG, +ICs and NBEs polarities

As shown in Figure R5b, the occurrence of negative NBEs is correlated with normal +ICs and not as similar as competition with positive NBEs. This may be because the conditions for the initiation of positive NBE are relatively strict. According to the interferometer observations reported by Lyu et al., (2018), few fast positive breakdowns (FPB) were observed in a thunderstorm, but more normal ICs pulses were observed.

3. You discuss the changes in the charge arrangement leading to the NBE polarity switching. It would be useful to formulate clearly what generally caused the observed enormous amount of NBEs and how often this can happen, especially if you want to use the presence of radio signals emitted by NBEs as an indicator of an exchange of greenhouse gases between the troposphere and stratosphere.

Response: This is a good comment. The tropical storm in our work is indeed a surprising case associated with an unprecedented number of NBEs. For the reason of the enormous amounts of NBEs. The radar and NBEs distributions suggest that the strong convection lift cloud above 20 km and the inferred upper positive charge region

is located near and above the tropopause, which is higher than the usual electrified thunderstorms reported in previous studies. The high altitude of the upper positive charge region may contribute to the outburst of negative NBEs. We suggest that the special meteorological and electrical structure with a high altitude of upper positive charge layer in the tropical storm of typhoon would provide favorable conditions for outbreak of negative NBEs.

The quantitative occurrence of NBEs requires statistical analysis. Following the referee's suggestion, we analyze the NBEs of thunderstorms observed in low-latitude at the GZ station for one year. Figure R6 shows the time series of hourly lightning and NBEs polarities counts. On average, the overall occurrence rates of NBEs are about 1.8% in low latitudes, which is higher than that of 0.5% in Great Plains thunderstorms as reported by Wiens et al., (2008).

Figure R6. Hourly lightning flashes and NBEs for the months of May–August 2019.

For the polarity, we see that NBEs are generally dominated with positive polarity in the low-latitudes thunderstorms except for some special days with outbreak of negative NBEs ($>10^2$ counts/hour). It is interesting to note that negative outbreaks are closely associated with tropical storms in typhoon, which is similar to the results in our work. Another question is raised: does outbreak of NBEs serve as the perfect proxy for typhoon development? It definitely merits further investigation in the future work.

We now added the occurrence frequency of enormous NBEs on Lines 116-117.

4. Methods: The estimation of the altitude of NBE is a crucial part of the work. I would suggest to evaluate more precisely the error in the altitude calculation.

Response: Thanks for the comment. To estimate the uncertainty in result of NBEs discharges height due to the inaccuracy in determining location and reflection pulses, we made the following test. For a typical NBE, up to 5 km and 1- μ s are intentionally added in 2-D location and reflection pulses. Under such conditions, Figure R7 shows

that no larger than 0.8 km variance from the original result. We now supplement the details of the error estimation in Method and in Supplementary information.

Figure R7. Variation of NBE discharge altitude with different location error and time error of reflected wave for nine NBEs.

Please see revised manuscript (Lines 315-319):

To estimate the error of the NBE height, we performed the following test. For a typical NBE, up to 5 km and 1- μs are intentionally added in 2-D location and reflection pulses. Under such condition, the results suggest that the maximum error is no larger than 0.8 km variance from the original altitudes shown in Extended Data Fig. 8.

5. Fig.4a: I do not understand why the NBEs appeared only in groups which are separated by approx. 1 min of nothing, this cannot be natural, please explain.

Response: We sincerely appreciate the referee's careful review to point out the issue. The 1-minute interval is because for NBEs we have only taken into account minutes info and ignored the seconds when we plotted the NBEs scatter plots. We have now corrected this in the new Figures 3, 4, S4, and S5.

6. I am quite confusing that you use in different figures different time intervals, if there is a reason for such approach, could you explain it properly?

Response: Thanks for the comment. The time interval of the radar and cloud condition analyses in different figures did not fully align because we want to present the different conditions of the outbreak and disappearance of negative NBEs: During the period in

Figure 3, the negative NBEs bursts are produced while the positive NBE rarely occur. The cloud analysis shows that the area of the Above-anvil cirrus plumes (AACP) is closely related to the frequency of the bursts of negative NBEs, suggesting the close connection between the negative NBE and AACP. Since the AACP is a product of strong convection injecting the cloud into the stratosphere, it indicates that the overshooting cloud is in the decaying stage at this time. The simultaneous radar observations in Figures 3c and 3d show that the heights of its 40 dBZ echo are relatively low, also suggesting the decaying stage of the cloud.

During the period in Figure 4, the negative NBEs suddenly disappear and the positive NBEs win the competition and occurs explosively. The cloud top temperature varies slightly while the radar observation shows a significant difference in that there is extreme updraft in the cloud, suggesting that the extremely convective updraft can inhibit the generation of the negative NBEs rather than promote it. The above analysis can provide an illustration of the competition of NBEs polarity in overshooting tops.

The simultaneous observations of the overlapping periods of the cloud and radar are added in the supplementary information.

Reviewer #3:

This is a well-written paper. I have a few comments that the authors need to consider and address in the revised version:

Response: Thanks for your positive comments. We have now revised the manuscript based on your suggestions. The detailed point responses are listed below.

1. Lines 83-83: Kindly reword this sentence, I recommend inserting: “helps estimate” after “...and the reflected ones”.

Response: Done.

Please see revised manuscript (Lines 83-85):

The time delay of the main pulses allows estimation of the source location and the delay between the main pulses and the reflected ones helps estimate the source height

2. Lines 85-86 and Figure 2a. It appears that a significant number of the negative NBEs are above the troposphere. The authors should note here that previous observations have also reported above-tropopause NBEs in “convective surges (plumes) overshooting the tropopause” such as Smith et al. (2004) doi:10.1029/2002RS002790 and Nag et al. (2010) doi:10.1029/2009JD012957.

Response: Thanks for pointing out this. We have now added previous observations from the literature.

Please see revised manuscript (Lines 91-94):

The sources of positive NBEs are primarily between 10 km and 17 km and most of the negative NBEs are above the tropopause, suggesting a normally electrified storm with the main negative and positive charge regions, which is consistent with prior reported above-tropopause NBEs observation.

3. Line 107: The acronym PDP and its meaning is not introduced before in the text.

Response: Thanks for the comment. we have now added the acronym PDP in main text.

Please see revised manuscript (Lines 106-109):

We determine the Proportion of the Dominant number of positive or negative Polarity to the total number (PDP), which is defined as the number of NBEs of the dominant polarity to the total number of NBEs.

4. Lines 109-111: It is difficult to understand the significance of this finding without further explanation of what PDP means and how values greater than 0.75 is suggestive of "strong competition".

Response: To clearly explain what *PDP* means, we now move the definition of *PDP* to the main text. The “Proportion of the Dominant number of positive or negative Polarity to the total number” (*PDP*) is used to quantify the relation between positive and negative NBEs:

$$PDP = \frac{\max(+NBE, -NBE)}{\Sigma(+NBE, -NBE)}$$

A larger *PDP* indicates a larger difference in the proportion of positive and negative NBEs, suggesting greater competition between positive and negative NBEs during a given period. We give an example: 10 NBEs, including 8 negative and 2 positive NBEs, are generated in a period. The value of *PDP* in this period is $PDP = \frac{\max(8,2)}{\Sigma(8,2)} = \frac{8}{10} = 0.8$.

5. Lines 196-197: Need to discuss the reasoning for this statement regarding the lower breakdown threshold. Is it because of changes in atmospheric composition, pressure, and temperature? Perhaps need a reference like Rioussset et al. (2020) doi.org/10.1016/j.icarus.2019.113506?

Response: This is a good comment. According to Figure 1 in Rioussset et al. (2020), the lower breakdown threshold is attributed to less pressure and humidity for higher altitudes. We have added the reference and supplemented the discussion in the text.

Please see revised manuscript (Lines 208-211):

The higher negative charge layer would narrow the gap with the upper positive charge layer, enhancing the ambient electric field. Moreover, the breakdown threshold is lower because of less pressure and humidity for higher altitudes, facilitating the initiation of positive NBEs.

6. Line 210-213: What about the occurrence of "regular" ICs between the main negative and positive charge regions. They would also play a role (perhaps a more dominant one) than the negative CGs in this charge balance? Please clarify here.

Response: The referee brings up an excellent point. Following the referee's suggestion, we analyzed the distributions of -CGs and +ICs with different polarities of NBEs within a 25×25 km grid in a given 5-min bin in a tropical storm. A total of 114 grids with negative NBEs and 104 grids with positive NBEs are determined. Figure R8 shows that an outbreak of negative NBEs is generally associated with frequent -CGs and 'regular' ICs. Moreover, it is interesting to note that the +IC seems to be more correlated with negative NBEs outbreak, suggesting ICs indeed play a more dominant role than -CGs in the charge balance for the cloud top discharges.

Figure R8. Relationship among -CG, +ICs and NBEs polarities

REVIEWERS' COMMENTS

Reviewer #2 (Remarks to the Author):

The authors took into account all of my comments, even if some of them induced quite extensive work including a search for additional similar storm examples. I appreciate the authors' efforts to improve the manuscript accordingly, I do not have additional requests and recommend the manuscript for publication.

Reviewer #3 (Remarks to the Author):

The authors have addressed all my comments adequately and I recommend this article for publication.

Additionally, in my opinion the authors have mostly addressed Reviewer 1's comments appropriately. Previous studies of different aspects of CIDs/NBEs have been published in Nature Communications (see references 5 and 6 in item 2 below). This paper discusses the issue of polarity of NBEs/CIDs in detail, which is not found in abundance in the scientific literature.

I suggest the following additional changes necessary to fully address Reviewer 1's comments.

1. Comment #1: In order to address this comment from Reviewer 1, I suggest the authors reference the paper by Williams (2006) on polarity asymmetry in lightning as one of the outstanding mysteries along with a brief sentence or two in the introduction.

Williams, E.R., (2006), Problems in lightning physics—the role of polarity asymmetry, Plasma Sources Sci. Technol. 15 (2006) S91–S108, doi:10.1088/0963-0252/15/2/S12.

2. Comment #4: In order to address Reviewer 1's comment I suggest that the authors introduce a brief paragraph in the introduction on the physics of NBE/CID production along with pertinent references. A few sentences such as those below in the introduction should suffice.

CIDs/NBEs have been described as a short-channel (few 100 m) “bouncing wave” phenomenon from observations and modified transmission-line modeling [1, 2, 3, 4] of their broadband (in the VLF-HF range) narrow bipolar pulse signatures and the accompanying fine structure. Using data from broadband interferometry, the channel production mechanism of NBEs has been associated with the so-called fast positive and negative breakdown in which streamers propagate at speeds of 10^7 - 10^8 m/s [5, 6, 7]. The exact nature of the fast breakdown that produces high-power (few tens of GW [4]) NBEs and can move at $>10^7$ m/s (up to 10^8 m/s, according to [7]) through virgin air remains a mystery. Fast breakdown could be a corona or ionization wave caused by positive corona from ice hydrometeors that is initiated ahead of the wave front, either in a forward or retrograde direction relative to the ambient electric field [7]. There are also recent attempts [e.g., 8] to model CIDs and explain their electromagnetic signatures that consider fast breakdown and the “rebounding” observed by Rison et al. [2016] using modified transmission line models.

1. Eack, K. B. (2004), Electrical characteristics of narrow bipolar events, *Geophys. Res. Lett.*, 31, L20102, doi:10.1029/2004GL021117.
2. Nag, A., and V. A. Rakov (2009), Electromagnetic pulses produced by bouncing-wave-type lightning discharges, *IEEE Trans. Electromagn. Compat.*, 50(03), 466–470.
3. Nag, A., and V. A. Rakov (2010a), Compact intracloud lightning discharges: 1. Mechanism of electromagnetic radiation and modeling, *J. Geophys. Res.*, 115, D20102, doi:10.1029/2010JD014235.
4. Nag, A., and V. A. Rakov (2010b), Compact intracloud lightning discharges: 2. Estimation of electrical parameters, *J. Geophys. Res.*, 115, D20103, doi:10.1029/2010JD014237.
5. Rison, W., P. R. Krehbiel, M. G. Stock, H. E. Edens, X. M. Shao, R. J. Thomas, M. A. Stanley, Y. Zhang (2016), Observations of narrow bipolar events reveal how lightning is initiated in thunderstorms, *Nature Communications* 7, Article number: 10721, doi:10.1038/ncomms10721.
6. Tilles, J., N. Liu, M.A. Stanley, P.R. Krehbiel, W. Rison, M.G. Stock, J.R. Dwyer, R. Brown, J. Wilson (2019), Fast negative breakdown in storms challenges electrical breakdown theory, *Nat. Commun* 10(1), 1648 (2019).
7. Krehbiel, P., C. da Silva, S. Cummer (2018), Continued mysteries of lightning studies, 16th International Conference on Atmospheric Electricity, Nara, Japan, 18-22, June, 2018.
8. Li, D., Luque, A., Gordillo-Vázquez, F. J., Silva, C. d., Krehbiel, P. R., Rachidi, F., & Rubinstein, M. (2022). Secondary fast breakdown in narrow bipolar events. *Geophysical Research Letters*, 49, e2021GL097452. <https://doi.org/10.1029/2021GL097452>.

Point-by-point Response to Reviewer's Comment

Manuscript number: NCOMMS-23-46957A

Manuscript Title: Polarity transitions of narrow bipolar events in thundercloud tops reaching the lower stratosphere

Reviewer #2:

The authors took into account all of my comments, even if some of them induced quite extensive work including a search for additional similar storm examples. I appreciate the authors' efforts to improve the manuscript accordingly, I do not have additional requests and recommend the manuscript for publication.

Response: Thanks for your constructive comments. We appreciate the reviewer's great efforts in improving this work.

Reviewer #3:

The authors have addressed all my comments adequately and I recommend this article for publication.

Response: We are grateful to the reviewer for the insightful comments, which has highly improved the quality of the manuscript.

Additionally, in my opinion the authors have mostly addressed Reviewer 1's comments appropriately. Previous studies of different aspects of CIDs/NBEs have been published in Nature Communications (see references 5 and 6 in item 2 below). This paper discusses the issue of polarity of NBEs/CIDs in detail, which is not found in abundance in the scientific literature.

I suggest the following additional changes necessary to fully address Reviewer 1's comments.

1. Comment #1: In order to address this comment from Reviewer 1, I suggest the authors reference the paper by Williams (2006) on polarity asymmetry in lightning as one of the outstanding mysteries along with a brief sentence or two in the introduction.
2. Comment #4: In order to address Reviewer 1's comment, I suggest that the authors introduce a brief paragraph in the introduction on the physics of NBE/CID production along with pertinent references. A few sentences such as those below in the introduction should suffice.

CIDs/NBEs have been described as a short-channel (few 100 m) "bouncing wave" phenomenon from observations and modified transmission-line modelin of their broadband (in the VLF-HF range) narrow bipolar pulse signatures and the accompanying fine structure. Using data from broadband interferometry, the channel production mechanism of NBEs has been associated with the so-called fast positive and

negative breakdown in which streamers propagate at speeds of 10^7 - 10^8 m/s. The exact nature of the fast breakdown that produces high-power (few tens of GW) NBEs and can move at $>10^7$ m/s (up to 10^8 m/s, according to) through virgin air remains a mystery. Fast breakdown could be a corona or ionization wave caused by positive corona from ice hydrometeors that is initiated ahead of the wave front, either in a forward or retrograde direction relative to the ambient electric field. There are also recent attempts to model CIDs and explain their electromagnetic signatures that consider fast breakdown and the “rebounding” observed by Rison et al. [2016] using modified transmission line models.

Response: Thanks for the constructive suggestion. The reviewer’s suggestion brings an excellent supplement to addressing the comment. We have now added a brief sentence on polarity asymmetry in lightning (Lines 65-66), and a brief paragraph in the introduction on the physics of NBE/CID production accordingly (Lines 56-63).